# Classification of knowledge of fertility period among adolescent girls in East Africa from 2012 to 2022: Machine learning algorithm

Andualem Addisu Birlie[1]*, Kassahun Dessie Gashu[2], Mulugeta Desalegn Kasaye[3], Ayana Alebachew Muluneh[3], Abdulaziz Kebede Kassaw[3], Hailemariam Kassahun Desalegn[3], Tamir Wondim Desta[4], Shimels Derso Kebede[3]

**1** Department of Health Informatics, College of Medicine and Health Science, Samara University, Samara, Ethiopia, **2** Department of Health Informatics, School of Public Health, College of Medicine and Health Science, Gondar University, Gondar, Ethiopia, **3** Department of Health Informatics, School of Public Health, College of Medicine and Health Science, Wollo University, Dessie, Ethiopia, **4** Department of Health Informatics, Arba Minch Health Science College, Arba Minch, Ethiopia

* me.yeadda@gmail.com

## Abstract

Understanding the time of the menstrual cycle would help women to avoid getting pregnant without the need for surgical, hormonal, or mechanical contraception. Women who do not use contraception and do not know when they are fertile are at a higher risk (17%) of unplanned pregnancy and abortion. Classifying knowledge of fertility periods using machine learning algorithms would help to automate decision-making, produce more precise and accurate classification, and scale up to manage big and complex datasets. Therefore, this study aimed to classify knowledge of the fertility period among adolescent girls in East Africa from 2012 to 2022 using a machine-learning algorithm. A community-based cross-sectional study design was used from 12 East African countries' DHS datasets spanning 2012–2022. The machine learning algorithms were applied to classify knowledge of the fertility period and identify its predictors using R software and Python, particularly Jupiter Notebook in Anaconda. Data cleaning, one-hot encoding, data splitting, data balancing, and ten-fold cross-validation were performed. Ten machine learning algorithms and SHAP were used to select and interpret the best model. From the 40,664 adolescent girls in East Africa, 13.22% (95% CI: 12.91, 13.54) of participants had knowledge of the fertility period. Logistic regression was found to be the best model for unbalanced training data with 74.38% of an AUC and 82.71% of an accuracy. While random forest outperformed on balanced training data, it achieved 91.12% of an AUC and 83.26% accuracy. The key determinant factors of the knowledge of the fertility period were education level, country, hearing about family planning, hearing about sexually transmitted infections, wealth index, knowledge of any method, and visiting health facilities. Governments, NGOs, policy makers, and researchers can utilize these

**Data availability statement:** The dataset used in this research is available from the DHS program official database (http://dhsprogram.com) upon formal request.

**Funding:** The author(s) received no specific funding for this work.

**Competing interests:** The authors have declared that no competing interest exist.

findings to design targeted interventions for improving adolescents' reproductive health based on the identified gaps and disparities.

## Author summary

The purpose of this study was to classify knowledge of the fertility period among adolescent girls in East Africa. Ten machine-learning models were trained, and the random forest model provided the most accurate prediction with an AUC score of 91.12% and an accuracy of 83.26%. Using the SHAP feature importance method, the top ten predictors were identified: education level, country, heard about family planning, heard about sexually transmitted infections, wealth index, knowledge of any method, and visited health facilities. Education level was a primary factor since girls with higher education levels were better prepared to understand reproductive health concepts. Similarly, increased exposure to hearing about family planning and general knowledge about STIs improved knowledge of fertility, again showing that a more extensive health communication strategy has contributed to girls' foundational knowledge. It is important to note that the wealth index also has an effect on access to media, education, and health services. We saw that girls who demonstrated that they were aware of a contraceptive method or had ever visited a health facility were significantly more likely to understand their fertility period.

## Background

According to the World Health Organization (WHO) definition, adolescent girls are women aged between 15 and 19 years [1]. Adolescent girls are at risk for serious health issues because they are going through a major physiological, psychological, and social transition from childhood to adulthood [2]. The menstrual cycle is a monthly natural change that occurs in the uterus and ovary of a mature woman, which has two subdivisions: the uterine cycle and the ovarian cycle [3]. Ovulation occurs around the middle of the menstrual period, approximately 14 days after the start of menstruation for a 28-day menstrual cycle [4].

Knowledge of the fertility period is understanding the possibility of getting pregnant during the menstrual cycle [5]. The women who were correctly knowledgeable about the fertility period were evaluated based on their perception that the fertility period occurs in the middle of the menstrual cycle [6]. Women who know their fertility periods are better equipped to make decisions on family planning, contraception, and general reproductive health as well as preventing unwanted pregnancies and terminating pregnancy [3,7].

Globally, approximately 16 million adolescent girls give birth yearly [8]. In Africa, the frequency of adolescent girls' pregnancy was 18.8%, with 19.3% of these cases occurring in Sub-Saharan Africa and 21.5% in Eastern Africa [9]. A significant portion of women in East Africa did not use contraception, which was one of the main

causes of the high fertility rates in the majority of East African countries, accounting for 4.78 children per woman [10]. The magnitude of unintended pregnancy among adolescent girls in East Africa was 54.6%, ranging from 36.15% in Rwanda to 65.29% in Zimbabwe [11]. Without the use of modern contraceptives, sexually active women who do not know their fertility period are at a higher risk (17%) of becoming pregnant unintentionally [7].

Despite the advantages of understanding one's fertility period, empirical data indicated that knowledge of the fertility period (KFP) was typically lacking globally, in Sub-Saharan Africa, and in East Africa [7,12]. This might be due to many people, families, and societies still viewing talking about fertility period issues as taboo [7,13]. The magnitude of KFP was 32.8% in the United States [3], 31.2% in Spain [14], and 15% in India [13]. The magnitude of KFP in Africa varied from 10.4% to 49%, 38.8% in West Africa [15], 15.5% in 29 African countries [15], and 24.04% in low-income African countries [16].

Women in the younger age groups who did not use contraception, did not know when they were fertile, and stopped using any form of modern contraceptive methods due to fear of the side effects were vulnerable to unwanted pregnancies, unsafe abortions, miscarriages, stillbirths, and maternal morbidity and mortality [17]. In addition, adolescent girls who did not use any contraception methods were vulnerable to school dropout, early marriage, early pregnancy, prolonged labor, gender inequality, illiteracy, unemployment, single motherhood, and other negative social outcomes [15]. The costs of the above-listed problems were reduced by adequate knowledge of the fertility period [18,19]. The determinants of KFP were residence, country, marital status, education level, employment status, wealth index, media exposure, visits by fieldworkers, contraception use, knowledge of any method, hearing about family planning, unmet need for contraception, and distance to health facilities [15,20].

However, many studies on reproductive health among reproductive-age groups, particularly adolescent girls, have primarily focused on the use of modern contraception, often overlooking the importance of fertility period knowledge in shaping reproductive health and broader health outcomes [21,22]. Consequently, as the number of input variables and potential correlations increases, traditional statistical methods have shown limitations in handling large and complex datasets [23]. Machine learning offers a practical solution to these challenges by effectively capturing intricate and nonlinear relationships within the data and supporting more robust, data-driven decision-making compared to earlier statistical approaches [24,25]. While logistic regression is effective for modeling linear relationships and interpreting variable significance, it assumes independence among predictors and may struggle with complex, nonlinear interactions to inform evidence-based decision-making [26]. Furthermore, many studies currently recommend artificial intelligence, data mining, and machine learning algorithms as ways to predict healthcare service delivery, such as the use of modern family planning, with different machine learning algorithms [27]. Machine learning algorithms help predict problems and their determinants using large data sets that assist public health decision-making research [28].

The purpose of this study was to generate new insights and classify adolescent girls' knowledge of the fertility period using advanced machine learning techniques and SHAP based feature importance analysis, drawing on the Demographic and Health Surveys (DHS) data from 12 East African countries spanning 2012–2022. Unlike previous research, which has not applied machine learning to this topic, the study incorporated a broader set of predictor variables including visits to health facilities, fertility preferences, breastfeeding status, history of terminated pregnancy or abortion, current abstinence, country of residence, unmet need for contraception, and amenorrhea status to enhance the depth and relevance of the analysis. The use of machine learning enabled the identification of key determinants and the development of predictive models that classify fertility period knowledge more effectively. This classification is vital for bridging gaps in awareness and beliefs, which in turn supports efforts to reduce financial barriers, maternal and child mortality, and societal pressures. The findings offer actionable guidance for policymakers and program managers, suggesting targeted interventions such as promoting awareness of family planning and sexually transmitted infections, strengthening health education, expanding access to quality health services, and refining health education policy frameworks.

## Methods

### Study design and study settings

The community-based cross-sectional study was conducted in 12 East African countries between 2012 and 2022. This study used standardized DHS data collected from 12 East African countries, namely Ethiopia (2016), Kenya (2022), Uganda (2016), Tanzania (2022), Burundi (2017), Rwanda (2019), Madagascar (2021), Mozambique (2022), Zimbabwe (2015), Zambia (2018), Malawi (2016), and Comoros (2012).

### Study population, data source and sampling procedures

All adolescent girls who have been in the selected enumeration areas and whose data were recorded in the dataset were included. In addition, adolescent girls with missing data values were excluded from the study. The Demographic and Health Survey (DHS) used stratified two-stage cluster sampling. DHS in the first stage created a region and in the second stage generated a sample of homes from each enumeration area. The knowledge of the ovulatory cycle (v217) variable was re-coded from the individual record (IR) dataset to produce the knowledge of the fertile period [29]. A total sample of 40,664 respondents was included in the study for further analysis. The complete process for determining the sample size was illustrated in Fig 1.

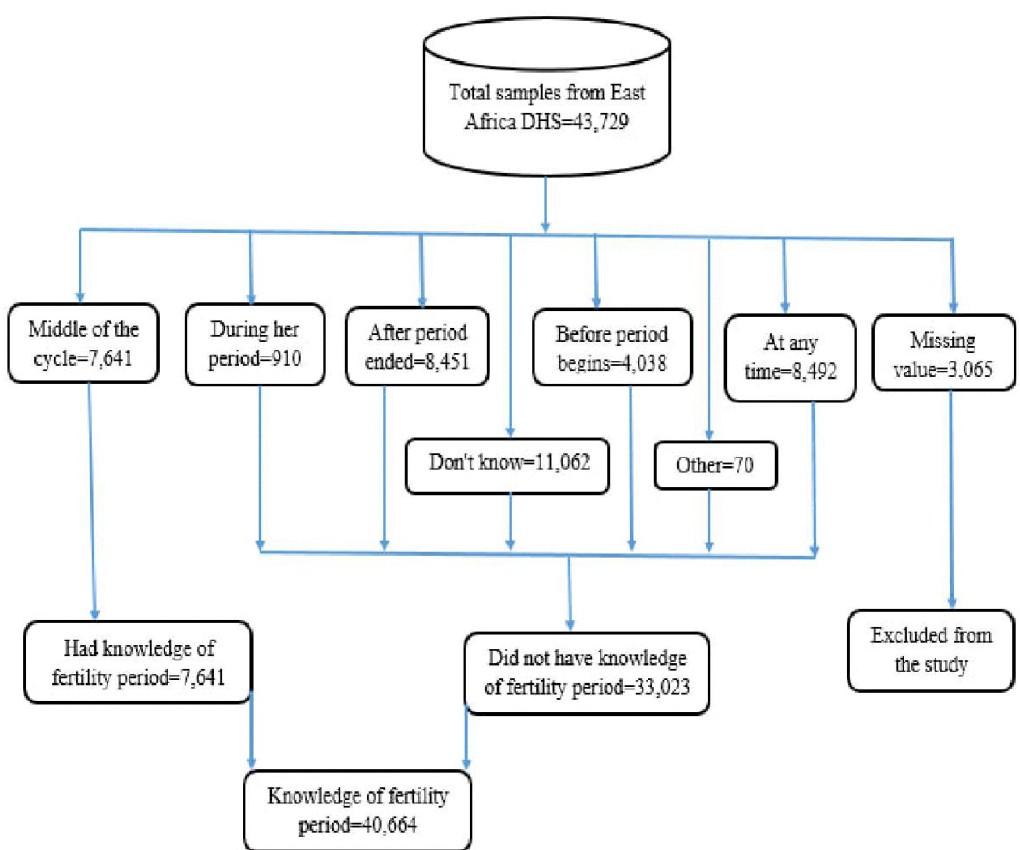

**Fig 1. Sample size determination of knowledge of fertility period.**

## Study variables

The dependent variable of this study was the knowledge of fertility periods, which was classified into two categories: good knowledge of fertility periods and poor knowledge of fertility periods [20,30]. The independent variables for this study were residence, country, marital status, education level, employment status, wealth index, media exposure, visited health facilities, heard about sexually transmitted infections, visited by fieldworker, contraception use, knowledge of any method. In addition heard about family planning, unmet need for contraception, distance to health facilities, currently pregnant, currently breastfeeding, currently amenorrheic, menstruation in the last 6 weeks, terminated pregnancy/abortion, recent sexual activity, fertility preference, and currently abstaining were independent variables for this study [15,20,30,31].

## Operational definition

**Adolescent girls.** Adolescent girls are young females who are found in the transitional stage between childhood and adulthood, usually between the ages of 15 and 19 [1].

**Good knowledge of the fertility period.** Knowledge of the fertility period was assessed by one standardized question: "When do you think a woman's fertility period is?" Respondents that answered "In the middle of the menstrual cycle" were classified as having good knowledge about the fertility period [20–22,32].

**Poor knowledge of the fertility period.** The same question, "When do you think a woman's fertility window is?" was used to assess poor awareness. Any respondent that answered "During her period," "After her period finished," or "Before her period started, at any time/course of a month," or "We don't know" reflects partial understanding; they do not accurately capture the biological cycle of peak fertility, which typically occurs around ovulation in the middle of the menstrual cycle and was categorized as having poor knowledge about fertility periods [20–22,30].

**Recent sexual activity.** A woman who experienced sexual intercourse within one month or four weeks [20].

## Data management and analysis procedures

Before developing a classification model, preprocessing the raw data for analysis was crucial to increase model accuracy and performance [33]. Yufeng Guo's 7 steps of machine learning were used to classify knowledge of fertility period, such as data collection, data preparation, model selection, model training, model evaluation, parameter tuning, and prediction [34]. In addition, interpretation of the model was performed using SHAP. STATA 17 and R 4.4.1 software were used for variable extraction and feature selection purposes, respectively. Python 3.10.2 using Jupyter Notebook with Scikit-learn, CatBoost, XGBoost, imbalanced-learn, Optuna, and SHAP packages was used to implement the machine learning algorithm technique. The analysis process summary was presented in Fig 2.

## Data source/collection

The data used in this study were obtained from the Demographic and Health Surveys (DHS) collected in 12 East African countries. Combined datasets from individual countries produced a sample of 40,664 adolescent girls for the final analytical set. This pooled dataset allowed for a broader assessment of reproductive health indicators across an array of regional contexts.

## Data preprocessing

Data preprocessing refers to the process of transforming raw, messy data to a clean, proper format that can be used for analysis, machine learning, or similar data-driven processes [35]. Data cleaning, class balancing, feature selection, and data splitting were among the data preparation techniques used in this study. These processes guarantee the data can be consistent, accurate, and suitable for modeling, which improves analytical algorithms' performance and reliability. Research shows that effective preprocessing can increase the accuracy of a model and reduce computational costs, making it a crucial step in any data-driven workflow [36].

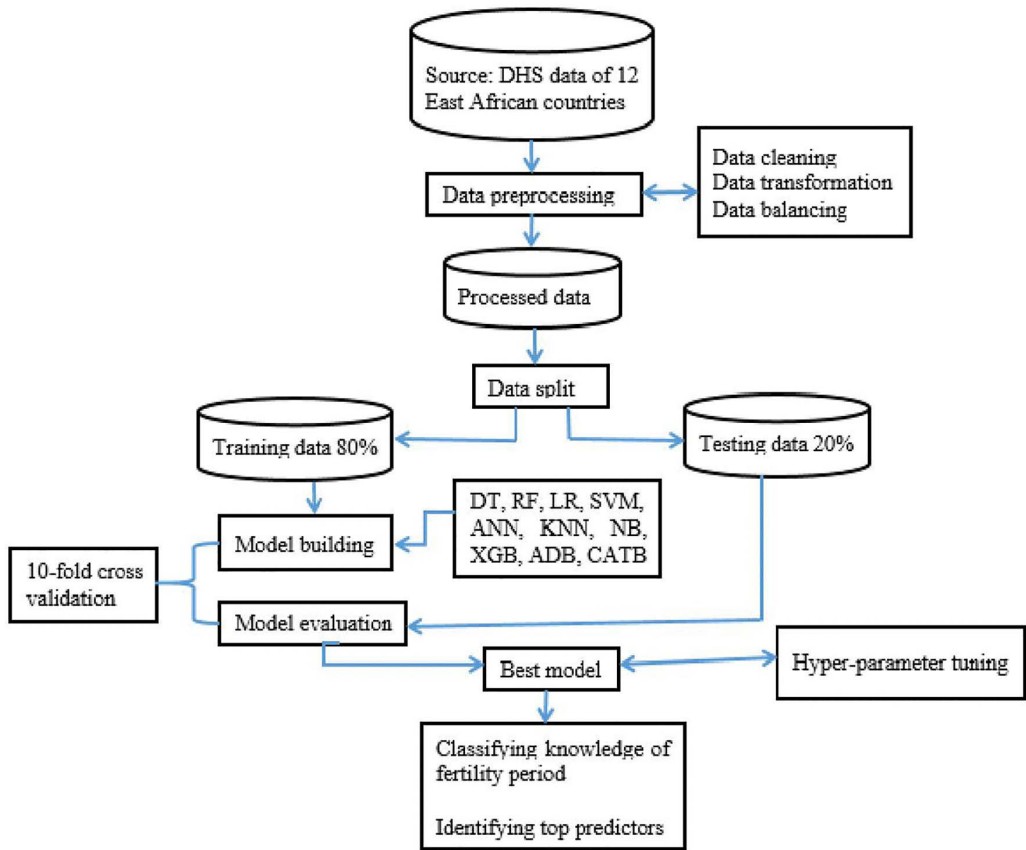

**Fig 2. Summery of data preparation and analysis process plan.** DHS = demographic health survey, DT = decision tree, RF = random forest, LR = logistic regression, ADB = adaptive boost, KNN = k-nearest neighbor, SVM = support vector machine, ANN = artificial neural network, NB = Naïve Bayes, XGB = extreme gradient boost, CATB = cat boost.

**Data cleaning.** Data cleaning was applied to address noise and missing values. It is impossible to send raw data through a training process and model testing since there might be missing data and the possibility of biased or misinformed results [37]. This dataset had no missing values, and missing value imputation techniques were not applied. The one-hot-encoding technique was used to encode categories to dummy variables [38]. Each categorical variable was coded as either "0" or "1" for the independent variables using the pandas get dummies technique. Features were transformed from multiple categorical values into discrete values like knowledge of fertility period, wealth index, and educational level.

**Class balancing.** A major problem in data mining and machine learning is class imbalance, which can lead to decreased classification accuracy, especially for the minority class [39]. When models are trained on imbalanced datasets, their performance generally gives priority to the majority class, leading to predictions that are biased and the minority-class instances being usually misclassified [40]. To address these issues, under-sampling, over-sampling, Synthetic Minority Over-sampling Technique (SMOTE), Adaptive Synthetic Sampling (ADASYN), and Synthetic Minority Over-sampling Technique Edited Nearest Neighbor (SMOTE-ENN) techniques were assessed [41]. Out of the tested resampling methods, under-sampling had the lowest performance and was omitted due to there being under-fitting. Although oversampling and the ADASYN technique performed better than under-sampling, their results were still much lower than those of SMOTE and SMOTE-ENN. Although SMOTE-ENN had the highest performance, it was eventually

dropped because of overfitting. Thus, SMOTE was selected as the best method for balancing the training dataset relating to knowledge of the fertility period among adolescent girls.

   **Feature selection.**  Feature selection is a method of dimension reduction, which entailed determining which independent variables were most pertinent and had the biggest influence on the target variable's classification [33]. In this study, feature selection was done using the Boruta algorithm. The Boruta algorithm determined which features in the dataset were highly and weakly relevant for the outcome variable. To select the features using the Boruta algorithm, we followed different steps, such as installing and loading required packages, preparing the dataset, running the Boruta algorithm, checking and plotting results, resolving tentative features, and extracting selected features. Finally, the whole data was split into training and testing by randomly assigning 80% of the data for model training and 20% for tuning the trained model.

## Model selection and development

Supervised classifier machine learning algorithms were used to evaluate the predictive power of machine learning techniques in classifying knowledge of fertility periods among adolescent girls. Decision trees, random forests, logistic regression, k-nearest neighbor, support vector machines, artificial neural networks, Naïve Bayes, adaptive boost, cat boost, and extreme gradient boost algorithms were used to select the best model. Those algorithms were chosen based on their interpretability, number of features, computational efficiency, accuracy, and characteristics of the dataset [42].

## Model training

The selected supervised machine learning classifiers were applied to analyze the dataset using a binary classification approach for knowledge of the fertility period. Once the models were chosen, training was conducted on both unbalanced and balanced versions of the dataset to evaluate performance. For the final classification on unseen test data, the best-performing predictive model was identified and trained using the balanced training dataset to ensure optimal generalization and accuracy.

## Model evaluation

The receiver operating characteristics (ROC) curve was used for evaluating a model's performance to distinguish between classes, selecting optimal thresholds, and comparing performances. Area under the curve (AUC), accuracy, sensitivity, specificity, positive predictive value (PPV), and negative predictive value (NPV) were used to assess the final model's performances. AUC was a reliable measure of performance, which showed the effectiveness of the model comparing adolescent girls who had knowledge of the fertility period and did not have knowledge of the fertility period. The higher AUC of the model on the balanced training dataset indicates the superior discriminatory performance compared to accuracy, sensitivity, specificity, PPV, and NPV [40].

   The model's performance can be further assessed using ten-fold cross-validation techniques in addition to the standard metrics. A ten-fold cross-validation entails splitting the data into ten subsets and training and evaluating the model ten times, each time using a different combination of nine subsets for training and one subset for evaluation [43].

## Hyper-parameter tuning

An external model setting known as a hyperparameter model requires the user to specify its value because it cannot be inferred from the data. The K-nearest neighbor algorithm's number of neighbors (K), which needed to be manually specified, was a basic example of a hyperparameter to improve and optimize the model's functionality [44]. Different kinds of hyper-parameter tuning techniques were used, like random search, grid search, and Bayesian optimization with the Optuna framework, to increase the performance of the model and select the best hyper-parameter tuning technique. Grid

search hyperparameter tuning was the best technique to produce the best model for this study when comparing with other techniques. In this study, the default setting performed better than the grid search. Because random forest is known for its low sensitivity to hyperparameters, especially in a balanced dataset [45]. Random forests typically allow trees to grow deep, which helps capture complex interactions. Tuning often restricts depth or splits to avoid overfitting, but in ensemble models, this can reduce diversity and weaken predictive [46].

### Making classification and model interpretation

Using certain predictor variables, the best-performing classifier with a certain accuracy was detected, whether adolescent girls had knowledge of the fertility period or not. Shapely additive explanations (SHAP) were used to interpret the model. Due to their "black box" character, powerful models (often tree-based models) were rarely explained or interpreted in machine learning research [47]. A new SHAP value analysis technique was used, which is based on game theory and can explain any machine learning model's classification, whether locally or globally, to reduce the limits on interpreting machine learning findings [48].

### Ethical consideration

This study utilized secondary data obtained from the East African countries' Demographic and Health Survey, which are publicly available through the Measure DHS program (www.measuredhs.com) upon formal request and legal registration. The dataset was accessed following approval from the Measure DHS program. Additionally, ethical clearance for the use of this data was obtained from the Ethical Review Board (ERB) of Wollo University, College of Medicine and Health Science (Ref Number: WUT.C.T/0025/13/2025).

## Results

### Characteristics of participants

**Socio-demographic and economic characteristics.** Among 40,664 adolescent girls, 28,399 (69.8%) were rural residents, and 19,145 (47.1%) were at the primary education level. The majority of adolescent girls, about 32,869 (80.8%), were not married, 26,993 (66.4%) were unemployed, and others were presented in Table 1.

The distribution of knowledge of the fertility period among adolescent girls across 12 East African countries was illustrated in Fig 3. Among these countries, Malawi contributed the largest share of participants, totaling 5,272 (12.9%) adolescent girls. In contrast, Comoros had the smallest representation, with 1,295 (3.18%) participants. This variation in sample size reflects differences in population coverage and survey scope across the included DHS datasets.

### Healthcare access and awareness characteristics

From the total participants, 36,148 (89.9%) used contraception, either modern or traditional; 37,725 (92.8%) had knowledge of any method; and 37,305 (91.1%) of adolescent girls had an unmet need for contraception, as presented in Table 2.

### Health service and awareness characteristics

Among the total participants, 38,549 (94.8%) of adolescent girls were not pregnant during the survey time, 36,025 (88.6%) were not currently breastfeeding, 38,128 (93.8%) were not currently amenorrheic, 31,973 (78.6%) had menstruation within the last six weeks, and 39,820 (97.9%) did not terminate pregnancy, as illustrated in Table 3.

### Pooled prevalence of knowledge of fertility period

The pooled prevalence of knowledge of the fertility period among adolescent girls in East Africa was 13.22% (95% CI: 12.91, 13.54), as shown in Fig 4.

**Table 1.  Socio-demographic and economic characteristics of respondents.**

| Variables | Categories | Weighted frequency | Percent |
|---|---|---|---|
| Residence | Urban | 12,265 | 30.2 |
|  | Rural | 28,399 | 69.8 |
| Education level | No education | 2,682 | 6.6 |
|  | Primary | 19,145 | 47.1 |
|  | Secondary | 18,443 | 45.3 |
|  | Higher | 394 | 1.0 |
| Marital status | Not married | 32,869 | 80.8 |
|  | Married | 7,795 | 19.2 |
| Employment status | Unemployed | 26,993 | 66.4 |
|  | Employed | 13,671 | 33.6 |
| Wealth index | Poor | 14,561 | 35.8 |
|  | Middle | 7,640 | 18.8 |
|  | Rich | 18,463 | 45.4 |
| Media access | No | 27,267 | 67.1 |
|  | Yes | 13,397 | 32.9 |

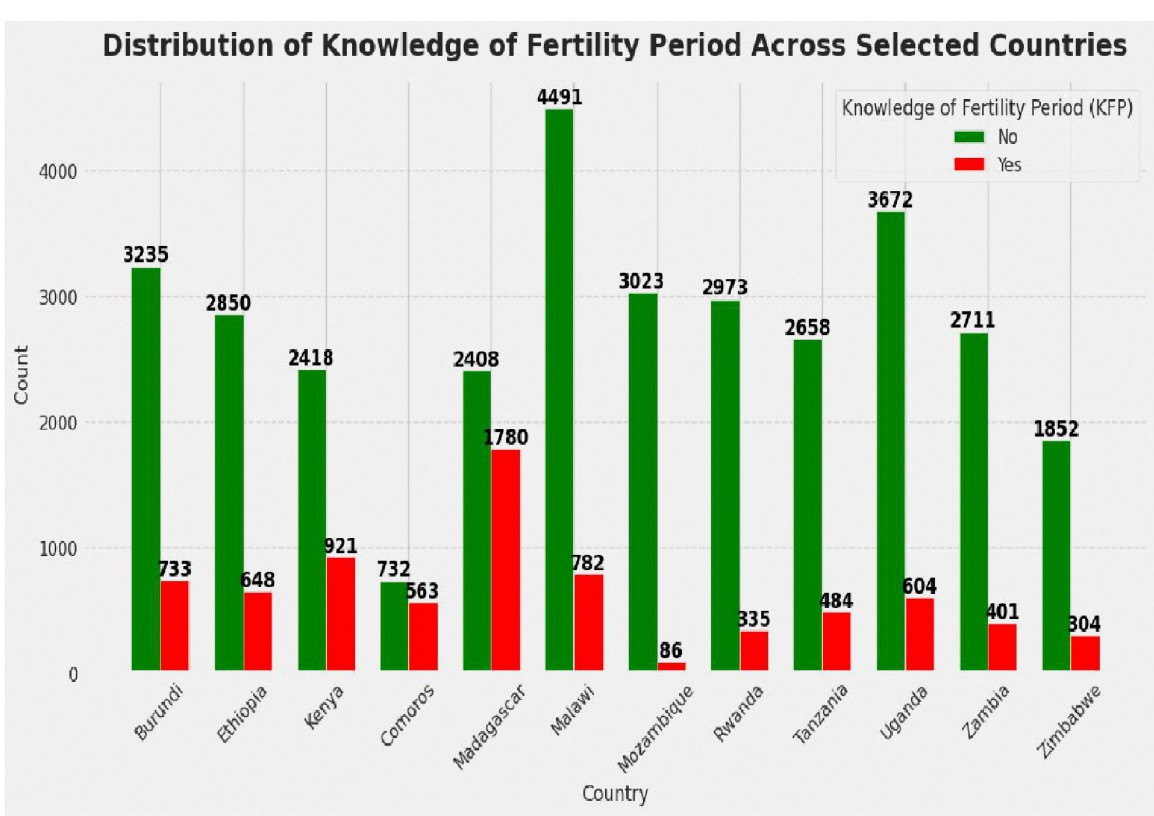

**Fig 3.  Distribution of knowledge of fertility period among adolescent girls across the East African countries.** No = poor knowledge of fertility period, Yes = good knowledge of fertility period.

**Table 2. Healthcare access and awareness characteristics of respondents.**

| Variables | Categories | Weighted frequency | Percent |
|---|---|---|---|
| Contraception use | No | 36,148 | 88.9 |
| | Yes | 4,516 | 11.1 |
| knowledge of any method | No | 2,939 | 7.2 |
| | Yes | 37,725 | 92.8 |
| Unmet need contraception | No | 37,305 | 91.7 |
| | Yes | 3,359 | 8.3 |
| Heard about family planning | No | 24,280 | 59.7 |
| | Yes | 16,384 | 40.3 |
| Visited by fieldworker | No | 37,098 | 91.2 |
| | Yes | 3,566 | 8.8 |
| Visited health facilities | No | 24,536 | 60.3 |
| | Yes | 16,128 | 39.7 |
| Heard about sexual transmitted infection | No | 4,869 | 12.0 |
| | Yes | 35,795 | 88.0 |
| Distance to health facilities | Not big problem | 26,695 | 65.6 |
| | A big problem | 13,969 | 34.4 |

**Table 3. Health service and awareness characteristics of respondents.**

| Variables | Categories | Weighted frequency | Percent |
|---|---|---|---|
| Currently pregnant | No | 38,549 | 94.8 |
| | Yes | 2,115 | 5.2 |
| Currently breast feeding | No | 36,025 | 88.6 |
| | Yes | 4,639 | 11.4 |
| Currently amenorrhea | No | 38,128 | 93.8 |
| | Yes | 2,536 | 6.2 |
| Menstruation in the last 6 weeks | No | 8,691 | 21.4 |
| | Yes | 31,973 | 78.6 |
| Terminate pregnancy | No | 39,820 | 97.9 |
| | Yes | 844 | 2.1 |
| Recent sexual activity | No | 33,355 | 82.0 |
| | Yes | 7,309 | 18.0 |
| Fertility preference | No | 5,517 | 13.6 |
| | Yes | 35,147 | 86.4 |
| Currently abstaining | No | 38,259 | 94.1 |
| | Yes | 2,405 | 5.9 |

## Classifying knowledge of fertility period

**Data balancing.** From the imbalanced training dataset, 33,023 (81.2%) adolescent girls had poor knowledge about the fertility period, while 7,641 (18.8%) had good knowledge about the fertility period. To help balance the sample, the SMOTE technique was implemented, synthesizing 18,777 observations for the minority class. Additionally, 6,605 cases from the majority class of adolescent girls with poor knowledge of the fertility period were randomly removed to help balance the

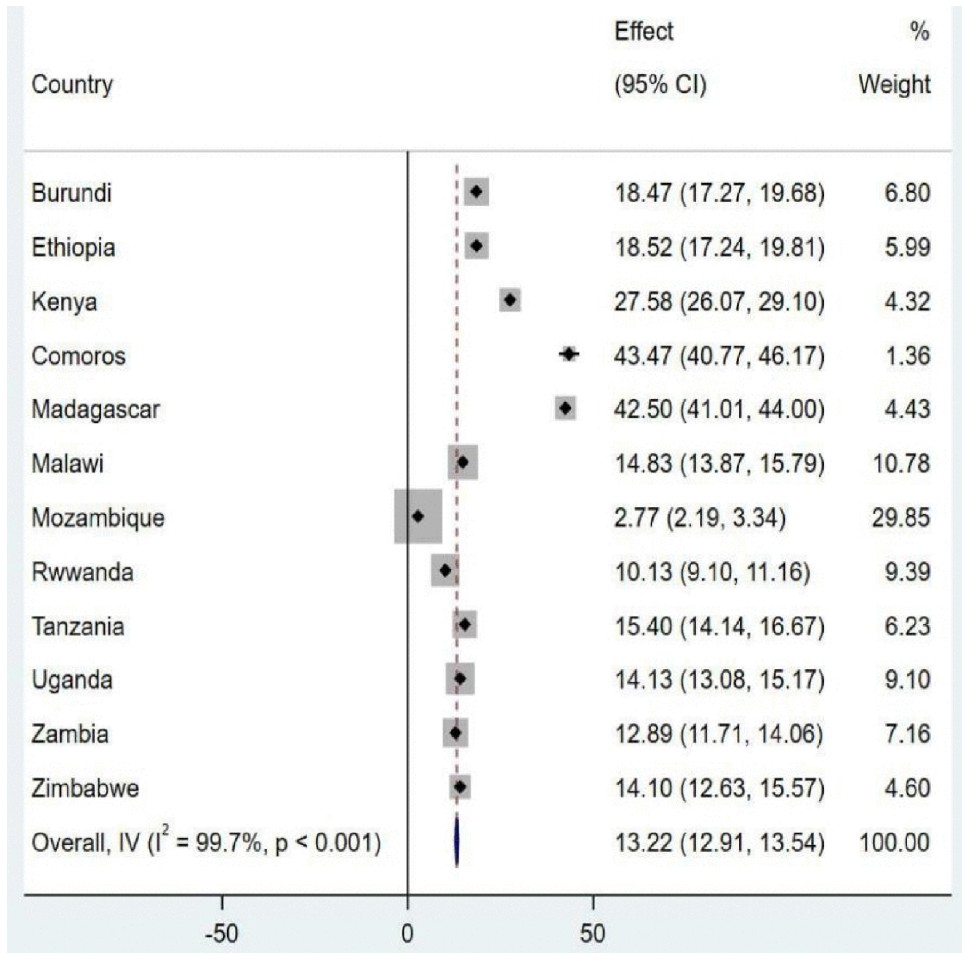

**Fig 4. The pooled prevalence of knowledge of fertility period.**

sample further. Thus, the final balanced training dataset consisted of 26,418 cases for both groups to had a 50/50% split among groups, as illustrated in Fig 5.

**Feature selection.** By using the Boruta feature selection technique, 22 variables were identified as important features out of the 23 variables for model development. Country, level of education, wealth index, marital status, heard about STI, breastfeeding, recent sexual activity, contraception use, knowledge of any method, current amenorrhea, residence, heard about family planning, currently abstaining were significant features. In addition unmet need for contraception, visited health facility, media exposure, distance to health facilities, current pregnancy, fertility preference, fieldworker visit, employment status, and terminating pregnancy/abortion were significant features to build the model and are indicated by the green color. Menstruation within the last six weeks was a non-significant variable and was excluded from the feature set, as indicated by the red color presented as shown in Fig 6.

**Model performance.** From the balanced training dataset, the random forest classifier algorithm was the best model, with a value of 91.12% of AUC, 83.26% of accuracy, 85.44% of sensitivity, 81.07% of specificity, 81.87% of PPV, and 84.77% of NPV, as shown in S1 Table.

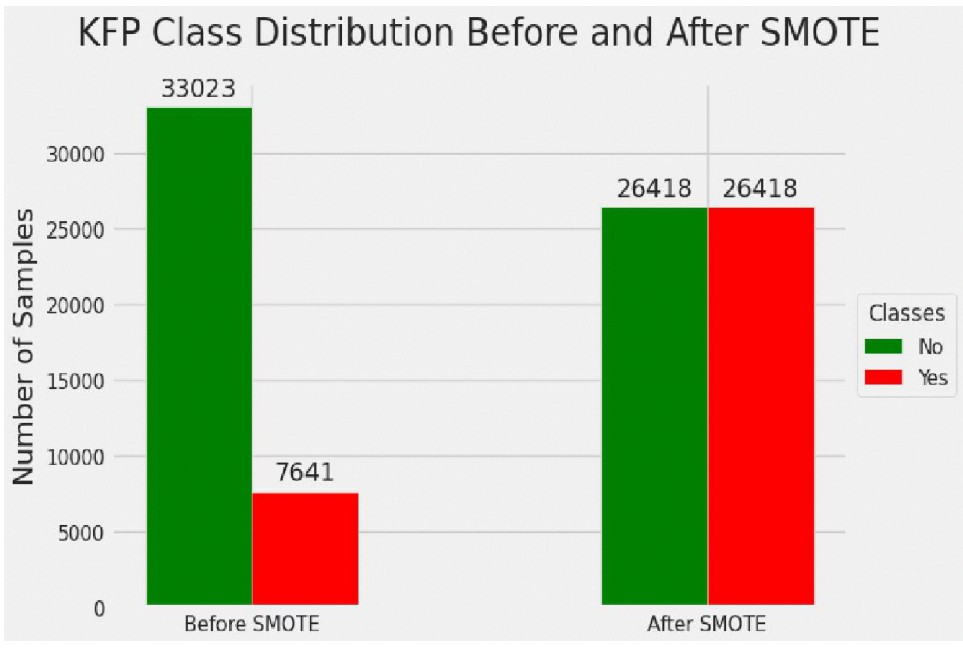

**Fig 5. Before and after balancing training data for knowledge of fertility period.** No = poor knowledge of fertility period, Yes = good knowledge of fertility period.

**ROC curve for the tested models.** Out of the ten machine learning algorithm classifiers used, the logistic regression model on the unbalanced training dataset and the random forest on the balanced training dataset had the highest AUC value on the ROC curve, as shown Fig 7.

**Hyper-parameter tuning.** The results of random search and Bayesian optimization with Optuna framework technique was less than the grid search technique. In addition, the result of the grid search technique was less than the default one or the balanced training dataset with all metrics of AUC, accuracy, sensitivity, specificity, PPV, and NPV, as indicated in Table 4.

The optimal and default values for maximum depth, number of estimators, minimum leaf node, minimum split, and maximum features of the random forest classifier algorithm using grid search was presented in Table 5.

**Comparison of the performance of the best model.** Based on the result given, the random forest classifier was the best model for this dataset. Its values were different in the unbalanced training dataset, balanced training dataset, and hyperparameter tuning with the grid search technique. The values of the random forest model in AUC were 68.4%, 91.1%, and 90.2% on the unbalanced training dataset, balanced training dataset, and hyperparameter tuning with the grid search technique, respectively, as presented in Fig 8. The best performance of this model, which we called the random forest classifier, was measured on the balanced training dataset technique, and SHAP was performed on this model.

**Global feature selection with SHAP.** SHAP global feature importance was applied using a random forest classifier to identify the most influential predictors of adolescent girls' knowledge of the fertility period. Each independent variable was evaluated based on its mean absolute SHAP value (MASHAPV), which measures the average contribution of that variable to the model's predictions. Variables were ranked in descending order of importance, with higher MASHAPV indicating greater influence. The top predictors included having a secondary education level and residing in Mozambique or Madagascar, which emerged as the strongest determinants of knowledge of fertility period. Additional important features included hearing about family planning, living in Rwanda, hearing about sexually transmitted infections, belonging

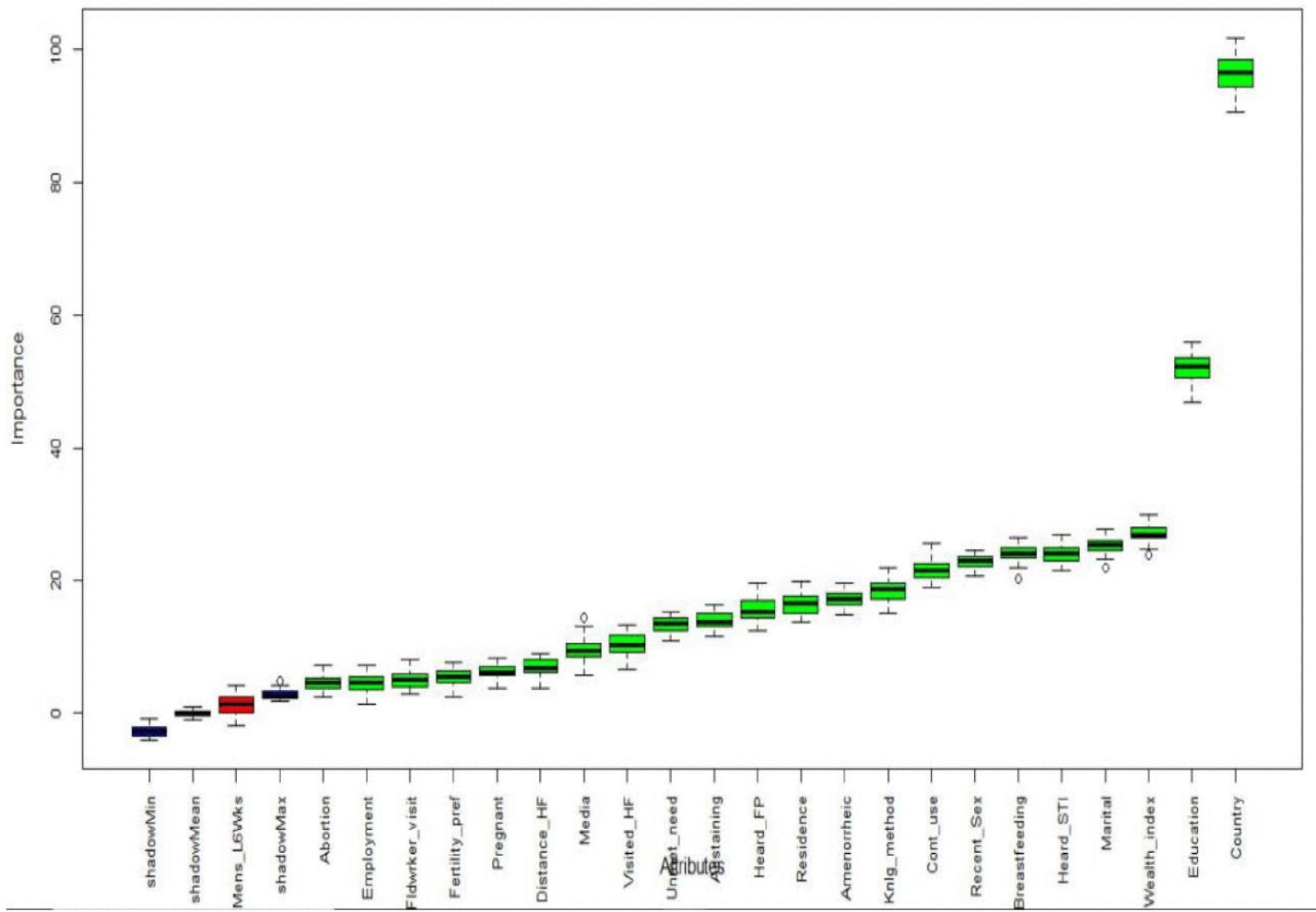

**Fig 6. Feature selection using Boruta algorithm.** Mens_L6Wks = menstruation within the last six weeks, Country = country, Education = level of education, Wealth_index = wealth index, Marital = marital status, Heard_STI = heard about STI, Breastfeeding = breast feeding, Recent_Sex = recent sexual activity, Cont_use = contraception use, Knlg_method = knowledge of any method, Amenorrheic = currently amenorrhea, Residence = residence, Heard_FP = heard about family planning, Abstaining = currently abstaining, Unmet_need = unmet need for contraception, Visited_HF = visited health facility, Media = media exposure, Distance_HF = distance to health facilities, Pregnant = currently pregnant, Fertility_pref = fertility preference, Fldwrker_visit = field-worker visit, Employment = employment status, and Abortion = terminating pregnancy/abortion.

to the rich wealth index, knowledge of any contraceptive method, visiting health facilities, and residing in Comoros were determinants of knowledge of fertility period, as illustrated in Fig 9.

**Model interpretation.** The beeswarm plot provides a compact visual summary of how each feature influences the model's prediction of whether adolescent girls have knowledge of their fertility period. Each dot represents an individual prediction, with color indicating the feature value (i.e., blue for low value and red for high value) and positions on the X-axis showing the direction and strength of impact. Dots on the right side (positive SHAP values) increase the likelihood of knowledge of the fertility period, while those on the left (negative SHAP values) decrease it.

In this model several features strongly increased the probability of knowledge of the fertility period; in Fig 10, adolescent girls who had secondary education, were from Madagascar or Comoros, had heard about family planning or sexually transmitted infections, were in the rich wealth index category, had knowledge of any contraceptive method, and had

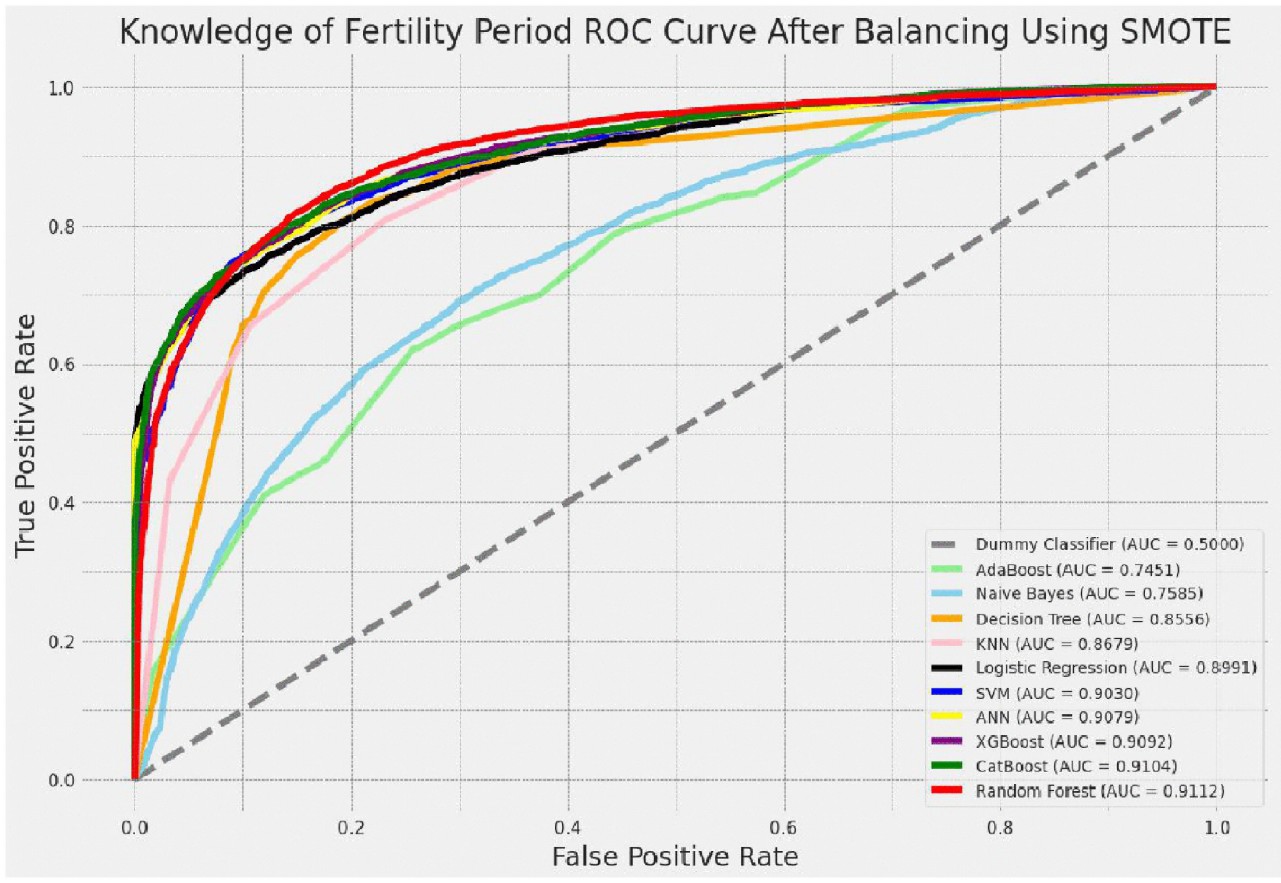

**Fig 7. ROC curve for after balancing using SMOTE resampling technique.** AUC = area under the curve, AdaBoost = adaptive boost, KNN = K-nearest neighbor, SVM = support vector machine, ANN = artificial neural network, XGBoost = extreme gradient boost, CatBoost = cat boost.

**Table 4. Comparison of different hyper-parameter tuning techniques.**

| Metrics | Hyper-parameter tuning | | | |
|---|---|---|---|---|
| | Default (%) | Tuned (%) | | |
| | | Random search | Grid search | Optuna framework |
| AUC | 91.12 | 88.93 | 90.22 | 90.09 |
| Accuracy | 83.26 | 80.15 | 81.59 | 81.45 |
| Sensitivity | 85.44 | 79.98 | 82.50 | 82.03 |
| Specificity | 81.07 | 80.32 | 80.68 | 80.86 |
| PPV | 81.87 | 80.25 | 81.03 | 81.08 |
| NPV | 84.77 | 80.05 | 82.17 | 82.82 |

visited health facilities. These factors reflect greater access to education, health services, and reproductive information. Conversely, being from Mozambique or Rwanda was associated with a decreased likelihood of knowledge of the fertility period, reflecting regional disparities in health education outreach or access. The beeswarm plot thus highlights both individual level and country level influences, offering actionable insights for targeted interventions in adolescent reproductive health programs.

**Table 5. Comparison of default and optimal values with grid search.**

| Grid search technique | Default | Optimal values |
|---|---|---|
| Number of trees (n_estimators) | 100 | 200 |
| Number of samples to draw from X to train each base estimator (max_depth) | None | 20 |
| Minimum number of samples required to split an internal node (min_samples_split) | 2 | 5 |
| Minimum number of samples required to be at a leaf anode (min_samples_leaf) | 1 | 1 |
| Number of features considered for the best split (max_features) | Square root | None |

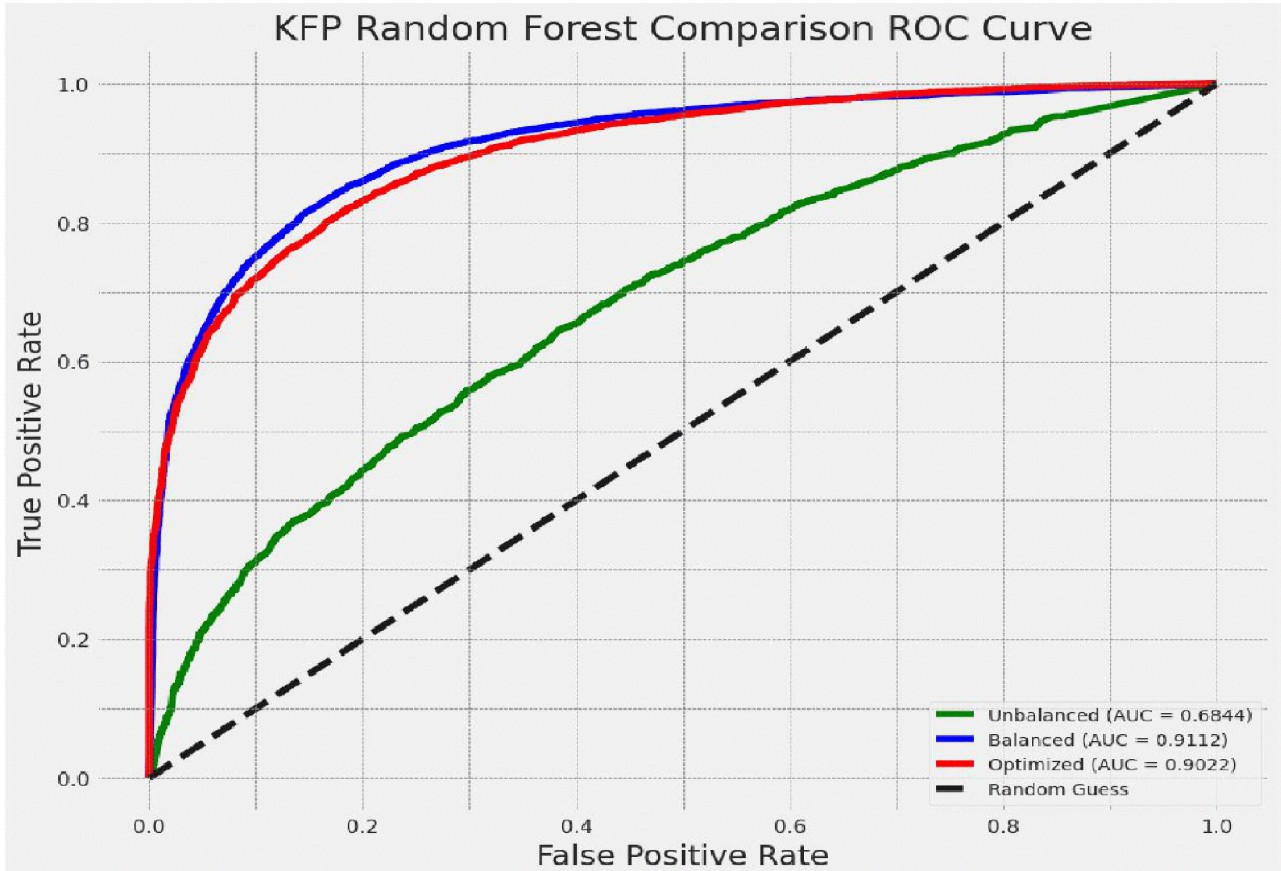

**Fig 8. Comparison of random forest model ROC curve.**

**Local feature importance with SHAP.** The SHAP waterfall plot in Fig 11 explains how individual features contributed to the model's prediction for one adolescent girl regarding her knowledge of the fertility period. The plot begins at the average model output (E[f(x)] = -1.577), which represents the baseline prediction before considering any personal characteristics. Each feature then pushes the prediction upward (red bars) or downward (blue bars) depending on whether it increases or decreases the likelihood of knowledge of the fertility period.

In this specific case, the final model output was f(x) = -1.873, meaning the model predicted that the adolescent girl did not have knowledge of her fertility period. This outcome was shaped by several negative influences: she had no media exposure, was not in the rich wealth category, had not heard about family planning, and was from Ethiopia but not

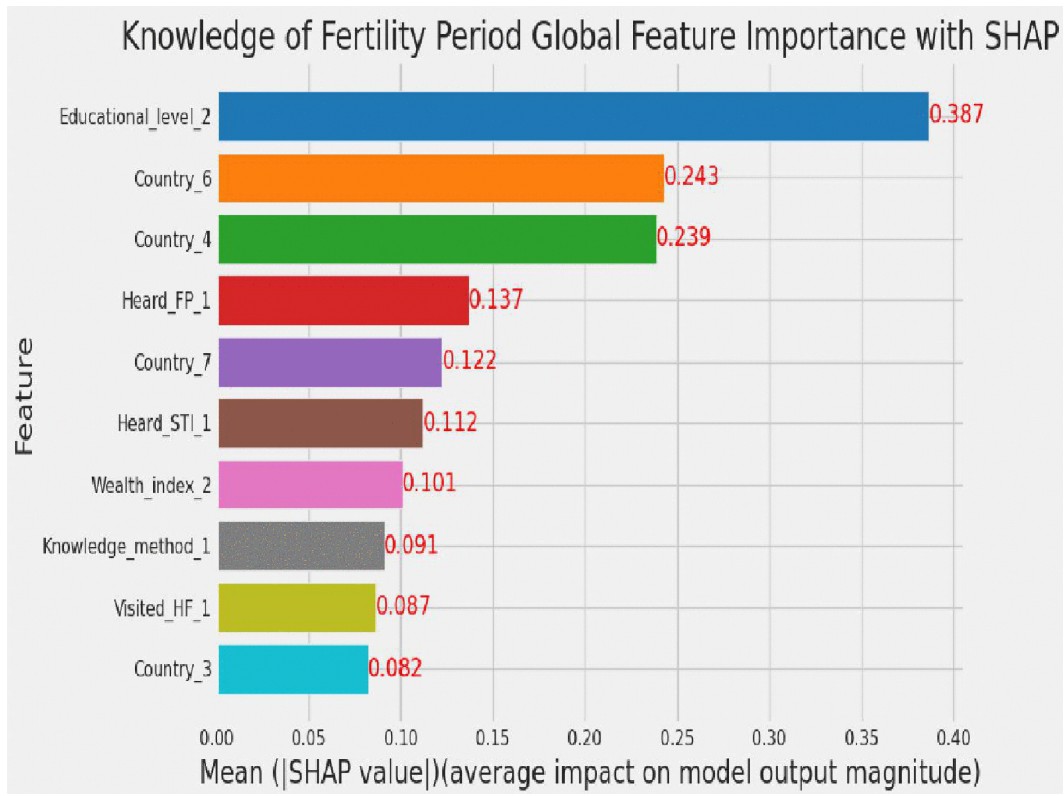

**Fig 9. SHAP global feature importance plot.** Educational_level_2 = secondary education level, Country_6 = Mozambique, Country_4 = Madagascar, Heard_FP_1 = heard about family planning, Country_7 = Rwanda, Heard_STI_1 = heard about sexual transmitted, Wealth_index_2 = rich wealth index, Knowledge_method_1 = had knowledge of any method, Visited_HF_1 = visited health facilities, Country_3 = Comoros.

Madagascar, all of which pulled the prediction downward. However, some features pushed the prediction upward: she had secondary education, had visited health facilities, was not from Mozambique, and had knowledge of contraceptive methods, indicating partial access to reproductive health resources.

## Discussion

This study aimed to classify knowledge of fertility period and its determinant factors among adolescent girls in East Africa using machine-learning algorithms. To conduct this research, ten machine-learning algorithms were used, such as random forest, decision tree, logistic regression, k-nearest neighbor, artificial neural network, support vector machine, Naïve Bayes, adaptive boost, cat boost, and extreme gradient boost, to get the best performance classifier model using both balanced and unbalanced training datasets. Data cleaning, one-hot-data encoding, 80/20% data split, and 10-fold stratified cross-validation were performed. Finally, the logistic regression classifier algorithm was the best model with the value of 74.38% of AUC and 82.71% of accuracy on the unbalanced training dataset. Whereas, from the balanced training dataset using the SMOTE resampling technique, the random forest classifier algorithm was the best model with a value of 91.12% for AUC and 83.26% for accuracy.

By using the random forest classifier algorithm with SHAP, the top ten determinant factors of knowledge of the fertility period among adolescent girls were identified. Those determinant factors were education level, country, hearing about family planning, hearing about sexually transmitted infections, wealth index, knowledge of any method, and visiting health facilities. These insights provide important information to program planners and health educators to help plan interventions

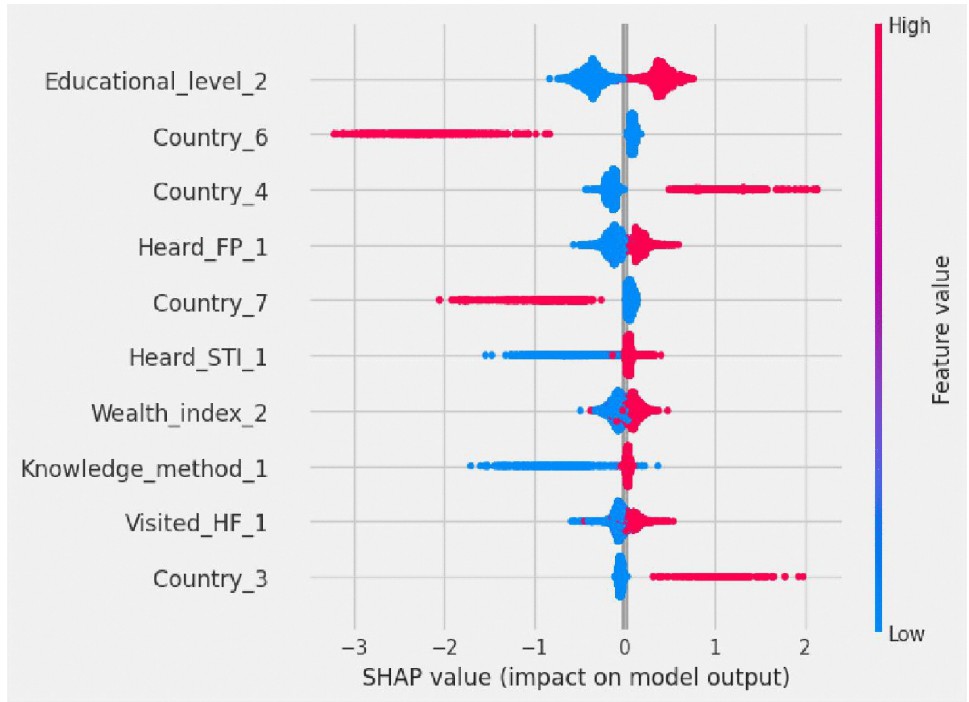

**Fig 10. Beeswarm plot ranked by mean absolute SHAP value.** Educational_level_2 = secondary education level, Country_6 = Mozambique, Country_4 = Madagascar, Heard_FP_1 = heard about family planning, Country_7 = Rwanda, Heard_STI_1 = heard about sexual transmitted, Wealth_index_2 = rich wealth index, Knowledge_method_1 = had knowledge of any method, Visited_HF_1 = visited health facilities, Country_3 = Comoros.

that are targeted, culturally relevant, and effective. With an understanding of who is informed, who had no access, and where the gaps are, program planners and health educators can better tailor messaging, allocate resources, and refine outreach to foster healthier behaviors and equitable access to care.

In this study, 13.22% (95% CI: 12.91, 13.54) of adolescent girls in East Africa had knowledge of the fertility period. It was less than the study conducted in low-income African countries (24.04%) [16], West Africa (38.8%) [15], and 29 African countries (15.5%) [15]. These differences might be due to the variation of study period and study design, quality of service utilization, and the size of the population included in the study [30]. In addition, it might be due to the disparity of social, cultural, economic, environmental, and healthcare access and educational factors across the world [16].

Adolescent girls who were in Mozambique and Rwanda had less likelihood of knowledge of the fertility period. However, those from Madagascar and Comoros had higher knowledge of the fertility period. The possible reason for this variation might be due to many factors, such as access to healthcare, education, cultural beliefs, and economic disparities, which contribute to country-level variations in fertility period knowledge [15]. Countries with stronger reproductive health education and awareness campaigns tend to have higher knowledge levels [13].

Adolescent girls who were at the secondary education level increased the likelihood of knowledge of the fertility period. This study was supported by findings conducted in 29 low-income countries [16], African countries [15], India [49], Ethiopia [20,32]. The reason might be due to the likelihood of increased knowledge of physiology or reproduction by women with higher education [20]. he other possible reason might be the students at the secondary educational level were more likely to encounter formal education on topics such as the menstrual cycle, fertility awareness, and family planning, especially in countries with strong curricula that integrated health education [16]. The presence of secondary education helped bridge gaps in understanding and can counter cultural taboos or misinformation, creating populations that were more informed [20,32].

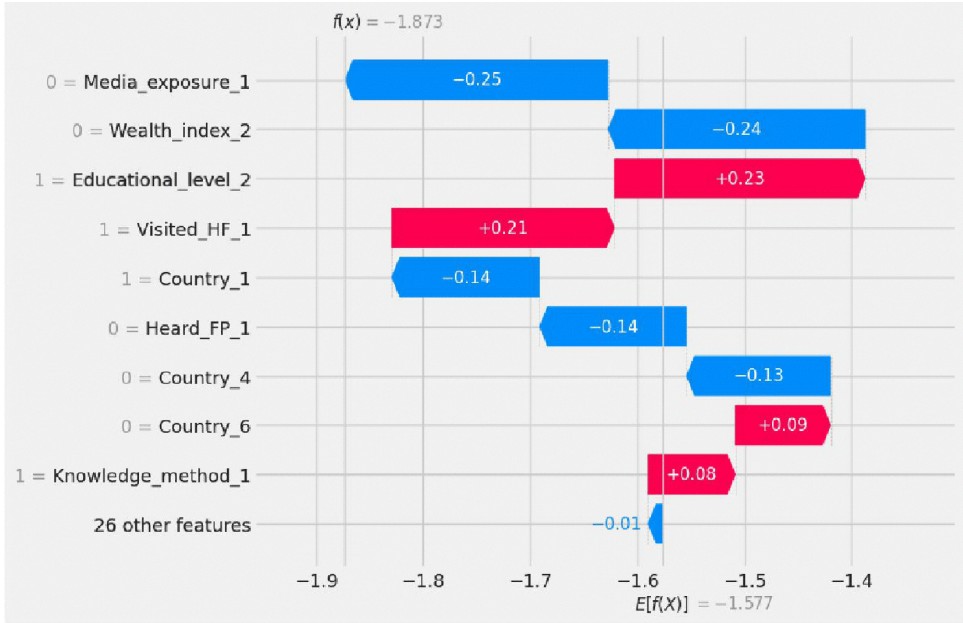

**Fig 11. Waterfall plot for secondary observation.** 0 = Media_exposure_1 = not media exposure, 0 = Wealth_index_2 = not being rich wealth index, 1 = Country_1 = being in Ethiopia, 0 = Heard_FP_1 = didn't heard about family planning, 0 = Country_4 = were not being in Madagascar, 1 = Educational_level_2 = being secondary education level, 1 = Visited_HF_1 = Visited Health facilities, 0 = Country_6 = not being in Mozambique and 1 = Knowledge_method_1 = had knowledge of any method.

Hearing about family planning increased the likelihood of knowledge about the fertility period. This finding was consistent with studies conducted in low-income African countries [16], 29 African countries [15], Kenya [30], and Ethiopia [32]. he possible reason might be due to family planning programs that emphasized understanding the menstrual cycle, ovulation, and safe periods as essential for effective contraceptive use [30]. Exposure to family planning messages through healthcare providers, media campaigns, or community outreach might enhance awareness and correct misconceptions about fertility [32].

Hearing about sexually transmitted infections increased the likelihood of knowledge about the fertility period. There was no study to support this finding. It might be due to STI education, which often included discussions about reproductive health and the menstrual cycle. Awareness campaigns and health programs addressing STIs typically emphasized the importance of understanding fertility to prevent complications like infertility, which could result from untreated infections [50].

Adolescent girls with a higher wealth index were more likely to have knowledge of the fertility period. This finding was consistent with studies conducted in low-income African countries [16], and Sub-Saharan Africa [31]. The possible reason might be due to the presence of better access to education, healthcare, and informational resources [31]. Wealthier households often prioritized education, including reproductive health topics, and had greater exposure to family planning programs and awareness campaigns [20]. Additionally, access to technology, such as the internet and media, enabled adolescents in wealthier settings to learn about fertility and reproductive health independently [16].

Adolescent girls who had knowledge about any contraceptive method increased the likelihood of knowledge of the fertility period. This finding was supported by studies conducted in low-income African countries [16] and India [49]. The possible reason for this finding might be knowledge of contraceptive methods helped individuals understand how to effectively plan their families, including identifying their fertile periods [49]. This knowledge empowers them to make informed decisions about when to conceive or avoid pregnancy [51].

Adolescent girls who visited health facilities were more likely to gain knowledge about the fertility period. There was no study to support this finding. The possible reason might be health facilities often provided access to reproductive health education, counseling, and family planning services [52]. Additionally, health facilities may offer informational materials and workshops that empower adolescents with accurate knowledge, helping them make informed decisions about their reproductive health [53].

**Strength and limitation of the study**

On the positive side, this study effectively leveraged machine-learning algorithms, which allowed for advanced and detailed classification of patterns related to knowledge of the fertility period. The use of the DHS dataset enhanced the reliability of findings, given its comprehensive nature and diverse demographic representation. Covering data from 2012 to 2022 also provided a decade-long perspective, capturing trends and variations over time. Furthermore, it focused on adolescent girls, addressing a critically important demographic in reproductive health, making its conclusions highly relevant to public health efforts.

However, the study faced certain limitations. It relied on self-reported data from the DHS dataset, which led to recall and social desirability biases. Some variables had high missing values, like husbands' educational level and self-reported status variables. The major limitation of the machine-learning algorithm for this study was that it was difficult to interpret using magnitude for the strength of association between the predictor variables with knowledge of the fertility period variable due to the absence of a regression coefficient.

**Conclusion**

This study used machine-learning algorithms to identify key factors influencing adolescent girls' knowledge of the fertility period in East Africa, revealing a low awareness rate of just 13.22%. Among ten models tested, the random forest classifier performed best on balanced data, identifying top predictors such as education level, country of residence, exposure to family planning and STI information, wealth index, knowledge of contraceptive methods, and visits to health facilities. These findings highlight significant disparities in reproductive health knowledge across socioeconomic and geographic lines, emphasizing the need for targeted, data-driven interventions.

Based on these insights, governments and NGOs should prioritize integrating comprehensive sexuality education into secondary school curricula, especially in countries with lower awareness levels like Mozambique and Rwanda. Programs should also expand youth-friendly services, enhance outreach through media and community platforms, and subsidize reproductive health services for low-income adolescents. Further research is needed to explore the impact of STI education on fertility awareness and to understand barriers to accessing health services. These actions can help bridge knowledge gaps, promote informed decision-making, and improve reproductive health outcomes among adolescent girls in the region.

Governments, NGOs, policy makers, and researchers can utilize these findings to design targeted interventions for improving adolescents' reproductive health based on the identified gaps and disparities. Education level could inform the development of age-appropriate, literacy-sensitive materials. Country-specific data will ensure culturally and policy-sensitive programming. If many adolescents report they have not heard about either family planning or STIs, then expanded outreach efforts through schools and the media, as well as peer education, would be warranted. The wealth index would assist in framing free or subsidized services and adopting a focus on low-income communities. Understanding knowledge of any method would also guide programs to reduce misinformation and promote a wider range of contraceptive options. Last, data on whether adolescents have ever visited health facilities would provide invaluable information on making youth-friendly services adequately available for care, reducing stigma around care, and making accessibility to care a priority. These findings also promote future research into exploring the effectiveness of current communication channels or barriers to accessing service among marginalized youth.

## Supporting information

**S1 Table. Comparison of default and optimal values with grid search.**
(DOCX)

## Acknowledgments

The authors acknowledge the Demographic Health Survey Program for providing the dataset and making the survey data accessible to the public for study.

## Author contributions

**Conceptualization:** Andualem Addisu Birlie, Kassahun Dessie Gashu, Mulugeta Desalegn Kasaye, Ayana Alebachew Muluneh, Abdulaziz Kebede Kassaw, Hailemariam Kassahun Desalegn, Tamir Wondim Desta, Shimels Derso Kebede.

**Data curation:** Andualem Addisu Birlie, Kassahun Dessie Gashu, Mulugeta Desalegn Kasaye, Ayana Alebachew Muluneh, Abdulaziz Kebede Kassaw, Hailemariam Kassahun Desalegn, Tamir Wondim Desta, Shimels Derso Kebede.

**Formal analysis:** Andualem Addisu Birlie.

**Investigation:** Andualem Addisu Birlie.

**Methodology:** Andualem Addisu Birlie.

**Resources:** Andualem Addisu Birlie.

**Software:** Andualem Addisu Birlie.

**Validation:** Andualem Addisu Birlie, Kassahun Dessie Gashu, Mulugeta Desalegn Kasaye, Ayana Alebachew Muluneh, Abdulaziz Kebede Kassaw, Hailemariam Kassahun Desalegn, Tamir Wondim Desta, Shimels Derso Kebede.

**Visualization:** Andualem Addisu Birlie.

**Writing – original draft:** Andualem Addisu Birlie.

**Writing – review & editing:** Andualem Addisu Birlie, Kassahun Dessie Gashu, Mulugeta Desalegn Kasaye, Ayana Alebachew Muluneh, Abdulaziz Kebede Kassaw, Hailemariam Kassahun Desalegn, Tamir Wondim Desta, Shimels Derso Kebede.

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
