## [Decision Letter · Decision Letter 0]

24 Sep 2025

Response to Reviewers
Revised Manuscript with Track Changes
Manuscript
**Journal Requirements:**

1. Please provide an Author Summary. This should appear in your manuscript between the Abstract (if applicable) and the Introduction, and should be 150–200 words long. The aim should be to make your findings accessible to a wide audience that includes both scientists and non-scientists. Sample summaries can be found on our website under Submission Guidelines: [LINK]

https://journals.plos.org/digitalhealth/s/submission-guidelines#loc-parts-of-a-submission

**Additional Editor Comments (if provided):**
**Reviewers' Comments:**

**Comments to the Author**

1. Does this manuscript meet PLOS Digital Health’s publication criteria?

Reviewer #1: Partly

2. Has the statistical analysis been performed appropriately and rigorously?

Reviewer #1: No

3. Have the authors made all data underlying the findings in their manuscript fully available (please refer to the Data Availability Statement at the start of the manuscript PDF file)?

Reviewer #1: Yes

4. Is the manuscript presented in an intelligible fashion and written in standard English?

Reviewer #1: No

Reviewer #1: Thank you for your submission. Your study addresses an important topic in adolescent reproductive health using modern machine learning methods. The use of DHS datasets from 12 East African countries adds strength to your analysis. However, the manuscript needs major revision to improve scientific clarity, transparency, and presentation. Below are my detailed comments:

1. Scientific Content and Justification

• You have applied machine learning (ML) to classify knowledge of the fertility period (KFP). However, you have not clearly explained why ML is needed over traditional statistical methods.

• Please explain how ML improves prediction or policy insights in this context. You may consider comparing with logistic regression outputs as a baseline.

2. Outcome Variable and Definitions

• You have defined "accurate knowledge" of the fertility period as only the response “middle of the cycle.” This is a narrow definition. Please justify this approach with references. Consider whether other responses could be partially correct.

3. Methodology and Model Transparency

• You have used 10 ML algorithms and SMOTE for class balancing. While this is good, the reason for choosing SMOTE over other techniques is not clearly explained.

• You should provide more details on:

o Feature selection steps (Boruta output)

o Data preprocessing

o Handling of missing data

o Hyperparameter tuning: why did default settings perform better than grid search or Optuna?

4. Performance Metrics

• You have reported accuracy and AUC but not F1-score, confusion matrix, or precision-recall. These are important, especially with imbalanced data.

• Please include confidence intervals for AUC and other metrics.

5. Interpretation of SHAP and Findings

• SHAP plots are a strength, but their interpretation is not clear. Please explain what the plots mean in simple terms and relate them to the real-world context.

• You list top features (e.g., education, wealth, country) but do not explain how these insights can help program planners or health educators.

6. Language and Grammar

• The manuscript contains many grammatical errors and awkward sentences. Examples include:

o "Random forest was emerged..." → should be "Random forest emerged..."

o "Bees warm plot" → should be "beeswarm plot"

• Please revise the entire manuscript carefully for grammar and clarity. A professional language edit is recommended.

7. Ethics and Data Availability

• You have clearly stated that DHS data was used and ethical clearance obtained. This is satisfactory.

• However, you should consider sharing your code or scripts (e.g., in a GitHub link) to improve transparency and reproducibility.

8. Conclusion and Implications

• Your conclusion repeats the results but does not offer clear recommendations for policy, programs, or further research.

• Please state how governments or NGOs can use your findings, especially in adolescent reproductive health programs.

• Some tables are too long and difficult to read.

• Please format tables clearly and ensure figure legends are complete and explained in the text.

**Do you want your identity to be public for this peer review?** For information about this choice, including consent withdrawal, please see our Privacy Policy

Reviewer #1: **Yes:** Dr K Madan Gopal

**Figure resubmission:**

**Reproducibility:** To enhance the reproducibility of your results, we recommend that authors of applicable studies deposit laboratory protocols in protocols.io, where a protocol can be assigned its own identifier (DOI) such that it can be cited independently in the future. Additionally, PLOS ONE offers an option to publish peer-reviewed clinical study protocols. Read more information on sharing protocols at https://plos.org/protocols?utm_medium=editorial-email&utm_source=authorletters&utm_campaign=protocols

---

## [Editor Report · Decision Letter 1]

5 Nov 2025

Response to Reviewers
Revised Manuscript with Track Changes
Manuscript
**Journal Requirements:**
**Additional Editor Comments (if provided):**
**Reviewers' Comments:**
**Figure resubmission:**

**Reproducibility:** To enhance the reproducibility of your results, we recommend that authors of applicable studies deposit laboratory protocols in protocols.io, where a protocol can be assigned its own identifier (DOI) such that it can be cited independently in the future. Additionally, PLOS ONE offers an option to publish peer-reviewed clinical study protocols. Read more information on sharing protocols at https://plos.org/protocols?utm_medium=editorial-email&utm_source=authorletters&utm_campaign=protocols

---

## [Editor Report · Decision Letter 2]

11 Nov 2025

Classification of knowledge of fertility period among adolescent girls in East Africa from 2012 to 2022: Machine learning algorithm.

PDIG-D-25-00447R2

Dear Birlie,

We are pleased to inform you that your manuscript 'Classification of knowledge of fertility period among adolescent girls in East Africa from 2012 to 2022: Machine learning algorithm.' has been provisionally accepted for publication in PLOS Digital Health.

Best regards,

Laura Sbaffi, PhD, MA, MSc

Section Editor

PLOS Digital Health